# Inference Scaling Laws:
# An Empirical Analysis of Compute-Optimal Inference for LLM Problem-Solving

**Yangzhen Wu**[1]*, **Zhiqing Sun**[2], **Shanda Li**[2], **Sean Welleck**[2], **Yiming Yang**[2]
[1]Institute for Interdisciplinary Information Sciences, Tsinghua University
[2]School of Computer Science, Carnegie Mellon University
wuyangch21@mails.tsinghua.edu.cn
{zhiqings, shandal, swelleck, yiming}@cs.cmu.edu

## Abstract

While the scaling laws of large language models (LLMs) training have been extensively studied, optimal inference configurations of LLMs remain underexplored. We study *inference scaling laws* and *compute-optimal inference*, focusing on the trade-offs between model sizes and generating additional tokens with different inference strategies. As a first step towards understanding and designing compute-optimal inference methods, we studied cost-performance trade-offs for inference strategies such as greedy search, majority voting, best-of-$n$, weighted voting, and two different tree search algorithms, using different model sizes and compute budgets. Our findings indicate smaller models (e.g., Llemma-7B) can outperform larger models given the same computation budgets, and that smaller models paired with advanced inference algorithms yield Pareto-optimal cost-performance trade-offs. For instance, the Llemma-7B model, equipped with our novel tree search algorithm, consistently outperforms Llemma-34B with standard majority voting on the MATH benchmark across all FLOPs budgets. We hope these findings contribute to a broader understanding of inference scaling laws for LLMs.[2]

## 1 Introduction

Scaling laws of neural networks [Hestness et al., 2017, Rosenfeld et al., 2019] have been established across a range of domains, including language modeling [Kaplan et al., 2020, Hoffmann et al., 2022, OpenAI, 2023], image modeling [Henighan et al., 2020, Yu et al., 2022, Peebles and Xie, 2023], video modeling [Brooks et al., 2024], reward modeling [Gao et al., 2023], and board games [Jones, 2021]. These studies have demonstrated how model performance is influenced by both the size of the model and the amount of training computation. However, there is limited knowledge on how varying the compute during *inference* affects model performance after the model has been trained.

To improve the task performance of large language models (LLMs), inference techniques typically involve additional computation as a *performance maximization* step at inference time [Nye et al., 2021, Wei et al., 2022, Wang et al., 2022b, Yao et al., 2023, Chen et al., 2024b]. The computational cost of these techniques must be taken into account for *compute-optimal inference*. For example, a Monte Carlo Tree Search (MCTS) method [Jones, 2021] may improve task performance, but potentially require much more compute than simply sampling solutions multiple times. Generally speaking, we need a comprehensive understanding of how various inference-time methods (e.g., best-of-$n$, majority voting [Wang et al., 2022a]) trade off between performance and cost. To improve

---

*Work done during the visit at Carnegie Mellon University.
[2]Project Page: https://thu-wyz.github.io/inference-scaling/

38th Conference on Neural Information Processing Systems (NeurIPS 2024) Workshop on MATH-AI.

our understanding, this paper presents a thorough empirical evaluation with careful analysis over various configurations of representative LLMs and inference algorithms.

Specifically, we explore how to select an optimal size for the language model and an effective inference strategy (e.g., greedy search, majority voting, best-of-$n$, weighted voting, and their tree-search variants) to maximize performance (i.e., accuracy) with a given compute budget. We control the inference computation (FLOPs) of a fixed model by generating more tokens through the language model[3], sampling further candidate solutions, and ranking them with a reward model. We analyze the performance of fine-tuned models of various sizes given different inference FLOPs on mathematical reasoning benchmarks (e.g., GSM8K test set [Cobbe et al., 2021a] and MATH500 test set [Hendrycks et al., 2021, Lightman et al., 2023b]). Our experiments cover several model families, including general-purpose LLMs (e.g., Pythia [Biderman et al., 2023] & Mistral [Jiang et al., 2023]) as well as math-specialized ones (e.g., Llemma [Azerbayev et al., 2023]).

Our results on Pythia (Fig. 1) illustrate how performance scales with increased inference compute across various model sizes. Typically, increasing the compute budget leads to higher accuracy until the accuracy reaches saturation. As the compute budget increases, smaller models initially perform better than larger ones, but once the accuracy of the smaller models saturates, the larger models have favorable performance. The right panel of Figure 1 demonstrates that the optimal model size for inference varies with different levels of computation. However, in real-world deployment, the available computation is typically much lower than the point where the accuracy of relatively small models saturates and larger models begin to show their advantage (as shown in Figure 2, where the 7B model outperforms the 34B model until 128 Llemma 7B solutions are sampled). This indicates that relatively smaller models could be more compute-optimal for inference.

We have also found that the commonly-used MCTS method does not perform well with weighted voting, as it often yields many unfinished solutions, hence having less effective votes. To address this issue, we propose a novel tree search algorithm, *REward BAlanced SEarch* (REBASE), which pairs well with weighted voting and achieves a Pareto-optimal trade-off between accuracy and inference compute. The key idea of REBASE is to use a node-quality reward to control node expansion, which eliminates the need for explicit rollouts while ensuring enough candidate solutions for voting.

In our experiments, REBASE consistently outperforms sampling and MCTS methods across all settings, models, and tasks. Importantly, we find that REBASE with a *smaller* language model typically achieves a Pareto-optimal trade-off. For instance, we show that the Llemma-7B model can achieve competitive accuracy to a Llemma-34B model while using $2\times$ less FLOPs when evaluating on MATH500 (Fig. 2) or GSM8K (Fig. 3). Moreover, Llemma-7B with REBASE outperforms Llemma-34B with standard majority voting across *all* compute budgets. Our results show the value of using smaller models with advanced inference-time algorithms, and the benefits of new algorithms for achieving better returns on inference-time compute.

## 1.1 Problem Formulation

We explore the following question: *Given a fixed FLOPs budget, how should one select an optimal model size for the policy model, and an effective inference strategy to maximize performance (i.e., accuracy)?* .

To address this, we represent the problem-solving error rate $E(N, T; \mathcal{S})$ as a function of the number of model parameters $N$, the number of generated tokens $T$ and the inference strategy $\mathcal{S}$. The computational budget $C$ is a deterministic function $\text{FLOPs}(N, T; \mathcal{S})$, based on $N$ and $T$. Our goal is to minimize $E$ under the test-time compute constraint $\text{FLOPs}(N, T, \mathcal{S}) = C$:

$$(N_{\text{opt}}(C), T_{\text{opt}}(C); \mathcal{S}) = \underset{(N, T, \mathcal{S}) \text{ s.t. } \text{FLOPs}(N, T, \mathcal{S}) = C}{\arg\min} E(N, T; \mathcal{S})$$

where $N_{\text{opt}}(C)$ and $T_{\text{opt}}(C)$ denote the optimal allocation of a computational budget $C$.

Here, the inference computation (FLOPs) for a fixed model can be modulated by generating more tokens with the policy model, e.g., by sampling additional candidate solutions and subsequently ranking them using a reward model. We primarily consider sampling and tree-search approaches with reranking or Majority Voting as the means to consume more tokens.

---

[3]Following Uesato et al. [2022], we refer to the main language model generating outputs as the *policy* model. It can be paired with a *reward* model, which scores outputs from the policy model to facilitate inference.

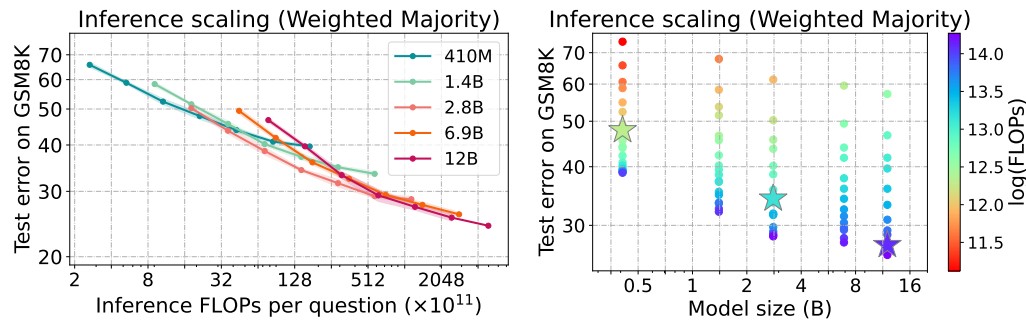

Figure 1: **The inference computation scaling laws** of Pythia exhibited in error rate on the **GSM8K** test set. We evaluate Pythia model using various sizes and various numbers of sampled solutions for majority voting. The *left* panel shows the error rate for each model size decreases steadily when the computation increases and converges at the end. The *right* panel shows the model performances given inference FLOPs budgets. In particular, the three stars highlight the optimal model size under $2^{41}$, $2^{44}$, and $2^{47}$ FLOPs, indicating that the optimal model size can vary given different budgets. Both the $x$ and $y$ axes are shown in log scale.

## 1.2 Inference Strategies

We examine sampling-based and tree-search inference strategies.

In sampling methods, majority voting (a.k.a. self-consistency [Wang et al., 2022a]) samples multiple reasoning paths, selecting the most frequent answer. When using a reward model, best-of-N selects the highest-scoring path, while weighted majority voting selects the answer with the highest weighted score, based on reward model values.

Tree-search methods, recently adapted for LLMs [Yao et al., 2023, Zhang et al., 2023, Zhou et al., 2023, Liu et al., 2024, Choi et al., 2023, Chen et al., 2024a, Tian et al., 2024, Chen et al., 2024a], often pair with value models to guide exploration. Monte Carlo Tree Search (MCTS) is common (Appendix C reviews MCTS), but it tends to be resource-heavy, requiring many more generated tokens than simpler methods. A more efficient inference strategy is needed, and comparisons of tree-search and sampling methods on computational cost are essential. We present our novel tree search method REBASE in 3.1.

## 2 Inference Scaling Laws

In order to compare the compute budgets of different models, we plot the figures with the number of FLOPs used per question during inference. We compute the inference FLOPs based on the standard formula from [Kaplan et al., 2020].

**Scaling law of compute-optimal inference for model size.** Figure 1 shows the relationship between inference compute and error rate for different model sizes. The error rate first decreases steadily and then starts to saturate. Initially, sampling many times from smaller models is compute-optimal. At larger compute budgets the larger models are preferable, since the performance of small models has saturated. As highlighted in the right panel of Figure 1, the optimal model size varies based on the inference budget. We performed a regression analysis on inference FLOPs $C$ and model sizes $N$ to establish a relationship between a given computational budget and its optimal model size. The resulting equation, $\log_{10}(C) = 1.19 \log_{10}(N) + 2.03$, lets us estimate the optimal inference model size for a specific compute budget.

**Llemma-7B model achieves competitive accuracy to Llemma-34B model with lower compute budget.** Fig. 2 & 3 shows the relationship between error rate and inference FLOPs for Llemma 7B and Llemma 34B using different inference strategies. Llemma-7B requires around $2\times$ less total FLOPs than Llemma-34B to achieve comparable accuracy. This held across inference strategies (sampling strategies, MCTS, REBASE) and tasks (MATH, GSM8K). This result suggests that, with the same training dataset and model family, generating more tokens with a suitable inference strategy using a smaller model can have more favorable cost-performance tradeoffs than using a larger model.

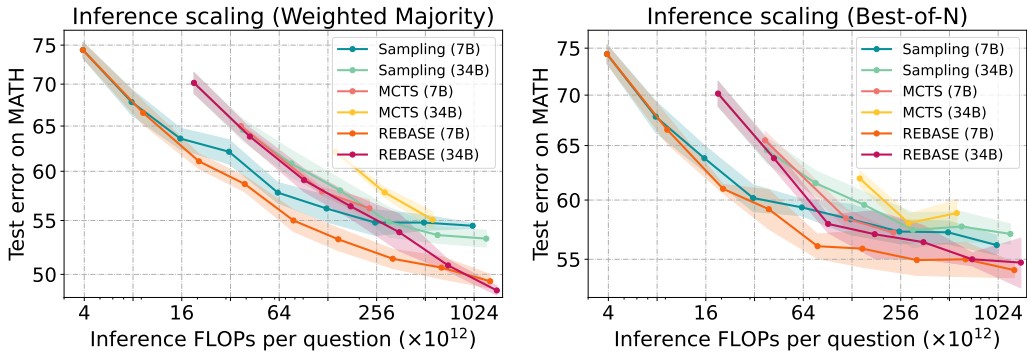

Figure 2: **The inference computation scaling comparisons across model sizes**. The left/right panel shows the problem-solving error rate on MATH based on Weighted Majority/Best-of-N.

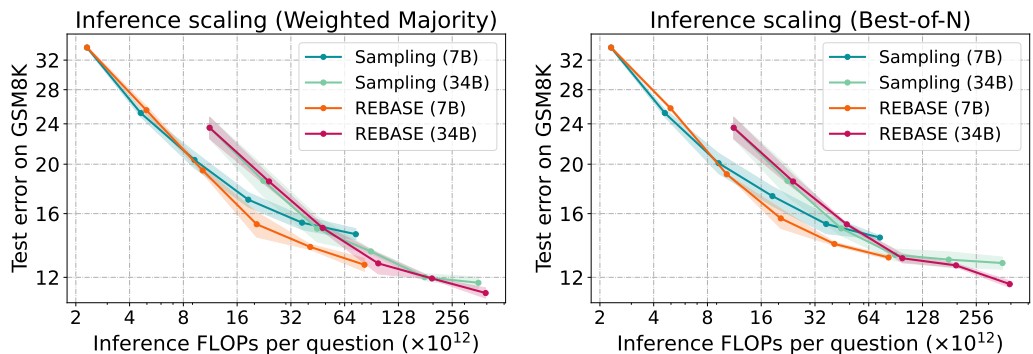

Figure 3: **The inference computation scaling comparisons across model sizes**. The left/right panel shows the problem-solving error rate on GSM8K based on Weighted Majority/Best-of-N. MCTS is not included in the comparison because of its poor compute-accuracy trade-off.

## 3 Compute-Optimal Inference

### 3.1 Reward Balanced Search (REBASE)

The REBASE tree search method inherits the exploitation and pruning properties of tree search, while using the reward model alone to estimate the nodes' qualities without additional computation for estimating values by sampling children. The details are provided below:

**Notations.** We view the fine-tuned LLM as a policy $\pi_\theta$ which generates the solution step by step. Given a question $x$ and the first $k$ steps of a solution $r_1 \cdots r_k$, the $(k+1)$-th step is sampled from $\pi_\theta(\cdot | xr_1 \cdots r_k)$. REBASE effectively generates a solution tree during inference, in which the root node the question $x$ and other nodes corresponds to solution steps. When generating solution trees, we generate children of $r_k$ by sampling from $\pi_\theta(\cdot | xr_1 \cdots r_k)$. Here we slightly abuse notations and use the corresponding question/solution step to denote a node. The reward of a node $r_k$ is generated by the PRM: $R(r_k) := R(qr_1 \cdots r_k)$.

**Initialization.** Given the question $x$, balance temperature $T_b > 0$, and sampling number of solutions $N$, we sample $N$ instances of the first step for the question, yielding all the nodes of depth 1 in the search tree. We let the sampling budget of depth 0, $B_0$, to $N$ at initialization.

**Reward modeling and update.** In the $i$-th iteration, the PRM assigns the rewards to all the nodes at depth $i$. After that, the algorithm examines whether the solutions up to depth $i$ are complete. Supposing there are $C_i$ completed solutions, we update the sampling budget using $B_i \leftarrow B_{i-1} - C_i$. If $B_i = 0$, the process ends, and we obtain $N$ solutions.

**Exploration balancing and expansion.** For all of the nodes $n_j$ with reward $R(n_j)$ in the depth $i$ of the tree, we calculate the expansion width of the $n_j$ as:

$$W_j = \text{Round}\left(B_i \frac{\exp\left(R(n_j)/T_b\right)}{\sum_k \exp\left(R(n_k)/T_b\right)}\right).$$

Then we sample $W_j$ children for $n_j$ for all the nodes in depth $i$, and start the next iteration.

Intuitively, when the balance temperature $T_b$ is small, this method encourages more exploitation which put much more compute budget on the nodes with high score, when $T_b$ is large, it encourages exploration where nodes with high score and low score are exlopred equally. In our experiment, we have found $T_b$ in the range of $(0.1, 0.3)$ works well for our process reward model.

## 3.2 Comparing REBASE to Other Baselines

**REBASE is Pareto-optimal.** REBASE consistently achieves the best cost-performance tradeoffs, outperforming the sampling-based methods in all settings when fixing the model and the evaluation task (Fig. 2, 3, 4, and 5). For example, in Figure 2, REBASE is the compute-optimal strategy at all inference compute budgets, with 7B typically the optimal model size. On the other hand, MCTS underperforms the sampling-based methods at each compute budget, likely due to its costly rollouts (Figure 2) compared to the efficient use of the reward model in REBASE.

Table 1 shows that REBASE achieves better accuracy with a lower compute budget compared to sampling-based weighted voting. With the 7B model, REBASE achieves higher accuracy with 7 times less compute. This finding is novel, and differs from previous tree search methods that typically improve the performance at the cost of higher computational expense compared to sampling-based voting [Chen et al., 2024a, Xie et al., 2023].

**Weaker models gain more from tree search.** For example, our proposed REBASE leads to $5.3\%$, $3.3\%$, and $2.6\%$ performance gains on MATH for Mistral-7B, Llemma-7B, Llemma-34B, respectively. The order of accuracy increase is inversely related to the model's corresponding greedy search accuracy on those datasets. This suggests that weaker models, as indicated by their lower greedy search accuracy, benefit more from tree search methods like REBASE.

**REBASE saturates later than sampling with higher accuray.** From Fig. 2 and Fig. 3, we observe that REBASE saturates later than the sampling methods, with lower final error rates. This is the evidence that REBASE improves the reasoning paths, due to the selection and pruning mechanism in the intermediate steps, REBASE discards the low quality partial paths and exploring more on good ones, results in a higher probability of sampling high-quality reasoning paths.

## 4 Conclusions and Limitations

**Conclusions.** In this work, we conducted a comprehensive empirical analysis of inference scaling law and compute-optimal inference for problem-solving with language models. We examined the scaling effect of computation during inference across different model sizes and found that while increased computation generally leads to higher performance, the optimal model size varies with the available compute budget. When the computation budget is limited, smaller models can be preferable. Additionally, we introduce our novel tree search method, REBASE, which is more compute-optimal than both sampling methods and Monte Carlo Tree Search (MCTS). REBASE typically achieves higher accuracy while using several times less computation than sampling methods. Our results underscore the potential of deploying smaller models equipped with sophisticated inference strategies like REBASE to enhance problem-solving accuracy while maintaining computational efficiency.

**Limitations.** Our empirical analysis specifically targets mathematical problem-solving. Investigating the inference scaling laws and compute-optimal inference strategies for tasks beyond mathematical problem-solving would be a valuable direction for future research. Additionally, we mainly evaluate the proposed REBASE on the GSM8K and MATH500 datasets. We speculate that the REBASE algorithm, which assumes access only to a function that assigns scores to nodes, will be effective in tasks beyond those studied here.

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

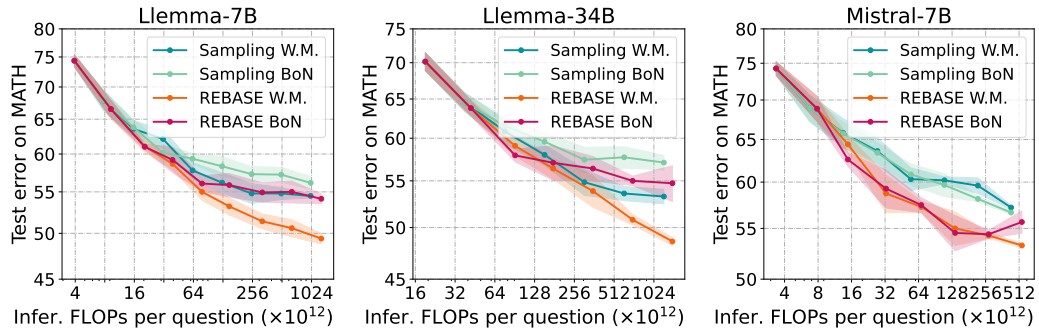

Figure 4: **MATH inference scaling across inference strategies and models** (lower is better). The tested models are Llemma-7B (left), Llemma-34B (middle), & Mistral-7B (right). In the legend, W.M. and BoN refer to weighted majority and best-of-$n$, respectively.

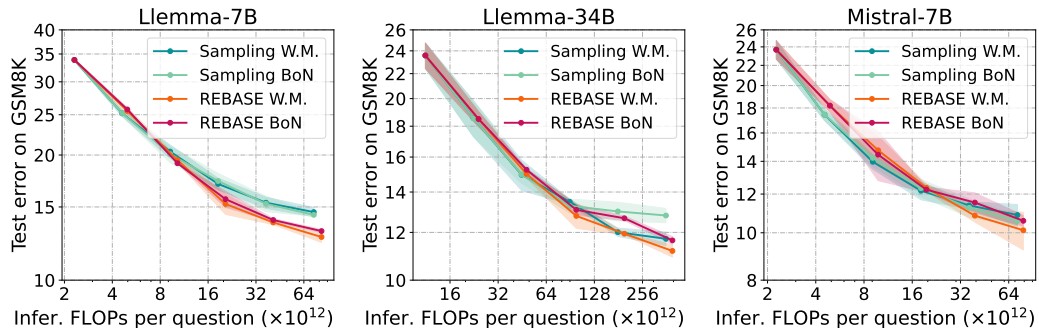

Figure 5: **GSM8K inference scaling across inference strategies and models** (lower is better). The tested models are Llemma-7B (left), Llemma-34B (middle), & Mistral-7B (right). In the legend, W.M. and BoN refer to weighted majority and best-of-$n$, respectively.

# A   Additional Results

In this section, we present the results of Llemma models and Mistral model to see how the REBASE performs on different models. Fig. 4 and 5 shows the scaling behaviors of Llemma models and Mistral model on MATH and GSM8k datasets. Table 1 shows the advantage of REBASE in specific compute settings.

Table 1: REBASE with a lower compute budget has better accuracy than sampling with a higher compute budget. We use weighted voting to aggregate candidates for both sampling and REBASE.

|  | # SAMPLES | FLOPS | MATH500 |
|---|---|---|---|
| MISTRAL-7B | | | |
| SAMPLING | 256 | $8.70 \times 10^{14}$ | 42.8 |
| REBASE | 32 | $\mathbf{1.36 \times 10^{14}}$ | **45.0** |
| LLEMMA-7B | | | |
| SAMPLING | 256 | $10.0 \times 10^{14}$ | 45.5 |
| REBASE | 32 | $\mathbf{1.48 \times 10^{14}}$ | **46.8** |
| LLEMMA-34B | | | |
| SAMPLING | 64 | $12.1 \times 10^{14}$ | 46.7 |
| REBASE | 32 | $\mathbf{7.08 \times 10^{14}}$ | **49.2** |

## B  Related Works

**Mathematical Reasoning with LLMs.**   Large language models have made significant progress in recent years, and have exhibited strong reasoning abilities [Brown et al., 2020, Hoffmann et al., 2022, Chowdhery et al., 2022, Lewkowycz et al., 2022]. Mathematical problem solving is a key task for measuring LLM reasoning abilities [Cobbe et al., 2021a, Hendrycks et al., 2021]. Ling et al. [2017] first developed the method of producing step by step solutions that lead to the final answer. Later, [Cobbe et al., 2021b] extended the work by training a verifier for evaluating and ranking solutions. Subsequent research (e.g., Lewkowycz et al. [2022]) has shown the performance benefits of inference-time techniques such as majority voting [Wang et al., 2022a] and weighted majority voting [Li et al., 2023]. We choose problem solving in mathematics as the task to study compute-optimal strategies since it allows us to accurately evaluate problem solving ability.

**Inference Strategies of LLM Problem Solving.**   A variety of inference strategies have been developed to generate sequences with a trained model. Deterministic methods such as greedy decoding and beam search [Teller, 2000, Graves, 2012] find highly probable sequences, often yielding high quality results but without diversity. Sampling algorithms (e.g., temperature sampling [Ackley et al., 1985]) can produce a diverse set of results which are then aggregated to achieve higher accuracy (e.g., via majority voting [Wang et al., 2022a]). Recent methods combine search algorithms with LLMs, including breadth-first or depth-first search [Yao et al., 2023], Monte-Carlo Tree Search (MCTS) [Zhang et al., 2023, Zhou et al., 2023, Liu et al., 2024, Choi et al., 2023], and guided beam search [Xie et al., 2023]. All of these methods show that using search at inference time can lead to performance gains in various tasks. However, the trade-off for the improved performance is the use of compute to perform the search. Analyzing the resulting cost-performance trade-offs remains understudied. In this paper, we systematically analyze the trade-off between compute budget and problem-solving performance, and propose a tree search method that is empirically Pareto-optimal.

**Process Reward Models.**   Process reward models (PRMs) have emerged as a technique to improve the reasoning and problem-solving capabilities of LLMs. These models assign rewards to the intermediate steps of the LLM generated sequences. PRMs have been shown effective in selecting reasoning traces with a low error rate, and for providing rewards in reinforcement learning-style algorithms [Uesato et al., 2022, Polu and Sutskever, 2020, Gudibande et al., 2023]. Ma et al. [2023] applies a PRM to give rewards on the intermediate steps and guide the multi-step reasoning process. The PRM can be either trained on human labeled data [Lightman et al., 2023a] or model-labeled synthetic data [Wang et al., 2023]. In our work, we use the PRM as the reward model for selecting generated solutions, and for selecting which partial solutions to explore in tree search.

## C  MCTS Details

In this section, we present additional background on the Monte Carlo Tree Search (MCTS) algorithm. The MCTS process can be formulated as the following steps:

**Selection.**   The process begins at the root node. Here, the algorithm recursively selects the child node that offers the highest Upper Confidence Bound applied to Trees (UCT) value, continuing until a node is reached that has not been expanded. The UCT is calculated using the formula

$$UCT(s) = Q(s) + C\sqrt{\frac{\ln\left(N(\text{Parent}(s))\right)}{N(s)}},$$

where $Q(s)$ denotes the quality score of node $s$, $N(s)$ is the number of visits to node $s$, $\text{Parent}(s)$ denotes the parent node of $s$, and $C$ is a constant determining the level of exploration.

**Expansion and evaluation.**   Upon reaching a non-terminal node $s$, the node is expanded by generating multiple child nodes. Each child node $c$ is then evaluated using a value function $V(c)$, which predicts the potential quality of continuing the sequence from node $c$.

**Backpropagation.**   After evaluation, the algorithm updates the UCT values and the visit counts for all nodes along the path from the selected node back to the root. For any node $n$ in this path, the

Table 2: Fine-tuning Hyper-parameters: LR refers to the learning rate, BS refers to the batch size. Pythia, Llemma-7B and LLemma-34B are the generators we use in our experiments, RM is short for Reward Model. We only use problems from GSM8K to train the Pythia models.

| Model | # Epoch | Dataset | BS | LR | Max Seq Length | Dtype |
|---|---|---|---|---|---|---|
| Pythia-410M | 1 | MetaMath (GSM8K) | 128 | 8E-5 | 768 | FP32 |
| Pythia-1.4B | 1 | MetaMath (GSM8K) | 128 | 4E-5 | 768 | FP32 |
| Pythia-2.8B | 1 | MetaMath (GSM8K) | 128 | 3E-5 | 768 | FP32 |
| Pythia-6.9B | 1 | MetaMath (GSM8K) | 128 | 2E-5 | 768 | FP32 |
| Pythia-12B | 1 | MetaMath (GSM8K) | 128 | 1E-5 | 768 | FP32 |
| Llemma-7B | 1 | MetaMath | 128 | 8E-6 | 1024 | FP32 |
| Llemma-34B | 1 | MetaMath | 128 | 8E-6 | 768 | FP32 |
| Llemma-34B RM | 2 | Math-Shepherd | 128 | 1E-5 | 768 | BF16 |

updates are made as follows:

$$N(n) \leftarrow N(n) + 1,$$
$$Q(n) \leftarrow \frac{(N(n) - 1) \, Q(n) + V(s)}{N(n)}.$$

## D  Hyper-parameters

**Finetuning**  All the hyperparameters for model fine-tuning can be found in Table 2. We preprocess the MetaMath [Yu et al., 2023] Dataset to make the solutions in a stepwise format.

**Inference**  For all the inference strategies, the temperature of the LLM is set to $1.0$. Max tokens for the output is $1024$ and max tokens for one step is $256$. For REBASE, we chose the balance temperature (the softmax temperature in the REBASE algorithm) as $T_b = 0.1$. For MCTS, we set $C$ in the UCT value as $1$ and we expand $4, 8, 16$ children for the root, $2$ children for other selected nodes with total $32, 64, 128$ expansions respectively.

