# OpenReview forum: "Inference Scaling Laws: An Empirical Analysis of Compute-Optimal Inference for LLM Problem-Solving"
_NeurIPS.cc/2024/Workshop/MATH-AI — MATH-AI 24_

### Official Review · Reviewer_s5dg · 2024-09-26
**Well-Executed Evaluation Framework with Opportunities for Further Exploration**

**Rating:** 7
**Confidence:** 3

**Review:**

# Pros
## 1. Strong Paper Writing, with Minor Typos:
The paper demonstrates good writing quality, although some typos are present, such as "Mjority" which should be corrected to "Majority." Please review the document for any additional typos.

## 2. Comprehensive Evaluation Settings:
The evaluation settings are thorough and well-detailed, covering aspects such as model size and inference FLOPs per question, which provides valuable insights.

# Cons
## 1. Need for a Theoretical and Detailed Scaling Formula:
While the paper does describe the scaling trends, I would prefer a more theoretical and detailed scaling formula to deepen the understanding of the trends observed.

## 2. Inclusion of Larger-Sized Language Models:
The largest model tested is LLemma-34B. Including experiments with even larger models could provide additional findings and more comprehensive insights.

## 3. Additional Benchmarks, e.g., AIME2024:
Adding more benchmarks, such as AIME2024, could help to further confirm whether the scaling law holds across different tasks, increasing confidence in the generality of the results.

---

### Official Review · Reviewer_pYq1 · 2024-10-06

**Rating:** 7
**Confidence:** 4

**Review:**

The paper investigates inference scaling laws of the task performances v.s. model parameters and inference FLOPs, and compare tree search variants to sampling-based methods. The paper finds that a smaller language model with a novel tree search algorithm typically achieves a Pareto-optimal tradeoff. The authors further propose REBASE, a tree search method that utilizes a separate reward model to determine the expansion of nodes. The empirical results show that REBASE performs better than other baselines (MCTS, sampling).

Pros:

- the paper provides solid empirical results to justify the advantages of the proposed REBASE method.

Cons:

- the setting of inference-time search often relies on a seperate reward model to aggregate the results of different trajectories (except for majority voting). The performance of REBASE would also likely to depend on the process reward model.

---

### Official Review · Reviewer_3Pvh · 2024-10-06
**This paper studies the scaling inference law for math problems solving with LLMs. They find that small model could have advantage when facing limited computation budget. To address the resource limit, they also propose a new algorithm REBASE and conduct numerical experiments to verify its performance.**

**Rating:** 6
**Confidence:** 3

**Review:**

This paper conducts a comprehensive empirical analysis of the scaling inference law and studies the inference strategies for solving math problems with Large Language Models. The authors investigate the selection of optimal model size and inference strategy under a fixed computational budget to maximize model performance. They discover that while increasing inference computation leads to better performance, the optimal model size varies depending on the available budget. Small models is likely to be more advantageous when the computation budget is limited.  To address the high cost of traditional MCTS, the authors propose a new Reward Balanced Search algorithm (REBASE). Numerical Experiments show that it is more computational efficient than MCTS and some other sampling methods.
Strengths:
1. By conducting extensive experiments across different model sizes and inference strategies, the authors reveal how inference computational budget affects model performance. The numerical experiment is convincing.
2. The intuition behind the new algorithm is simple but innovative by just replacing MCTS with some novel tree search method, but it remains good performance especially when per question Inference FLOPs is large
3. Also the study highlights the advantage of combining small models with REBASE when computational budget is limited, which sheds lights on how to deploying LLMs in resource-limited environments.
Weakness:
1. Lack of theoretical analysis and intuition. If the author could give us some mathematical intuition about the new algorithm design, then the paper could be more convincing.

2. I suggest the author comparing their method with more algorithms to better illustrate the power of the REBASE algorithm.

Questions: I am not an expert in LLM and empirical study, so please forgive me if my questions are naive.

1. Could you please provide some theoretical intuition about why REBASE works better? For example, why do you choose that exponential weighting:

$W_j=\mbox{Round}(B_i\frac{\exp(R(n_j)/T_b)}{\sum_k\exp(R(n_k)/T_b)})?$

2. The mathematical problem-solving tasks are very interesting. Are there any possibilities that your strategy could be applied to other types of problems such as natural language tasks? In other words, I’m curious that are there any special structures of mathematical problem-solving tasks that make your innovative method works well?

---

### Official Review · Reviewer_pqbc · 2024-10-07
**This paper investigates inference scaling laws for large language models (LLMs) in problem-solving tasks. It introduces the concept of compute-optimal inference, exploring trade-offs between model size and inference strategies. The authors present REBASE, a tree search method that outperforms sampling and MCTS in terms of compute efficiency. Empirical results on mathematical reasoning benchmarks demonstrate that smaller models with sophisticated inference strategies can achieve competitive performance while maintaining computational efficiency.**

**Rating:** 7
**Confidence:** 3

**Review:**

**Quality**

Strengths:
- Comprehensive analysis of inference scaling laws across different model sizes and compute budgets.
- Introduction of REBASE, the tree search method with clear explanation and empirical validation.
- Well-structured mathematical foundation for REBASE, using a softmax-like formula for exploration balancing.
- Empirical evidence showing REBASE's superior performance and compute efficiency compared to MCTS and sampling methods.

Weaknesses:
- The paper lacks a detailed discussion of limitations or potential drawbacks of the proposed method.
- No theoretical analysis of REBASE's convergence properties or regret bounds.
- Limited discussion on hyperparameter sensitivity, particularly the balance temperature T_b.
- Absence of comparative analysis with tree search variants beyond MCTS (e.g., beam search, A* search).
- No explicit discussion on how REBASE handles reward normalization across different tree depths.
- Lack of analysis on REBASE's performance across problems with different structures (e.g., sparse rewards, deceptive rewards).
- No discussion on memory requirements for storing the search tree, which could be a limiting factor for very large trees.

**Clarity of writing**

Strengths:
- Clear problem formulation and explanation of the research question.
- Well-structured explanation of the REBASE algorithm.

Weaknesses:
- The introduction could benefit from a more comprehensive review of related work.
- Lack of explicit discussion on potential limitations of REBASE and areas for future work.

**Originality of the paper**

Strengths:
- Novel exploration of inference scaling laws for LLMs in problem-solving tasks.
- REBASE's unique approach of balancing exploitation and exploration without additional value estimation computations.

Weaknesses:
- While REBASE is novel, it builds on existing concepts (e.g., tree search, sampling methods). The paper could benefit from a more detailed comparison with other advanced tree search methods.
- The reward-based node evaluation in REBASE is similar to some existing reinforcement learning techniques, which could be acknowledged more explicitly.

**Significance of this work**

Strengths:
- Implications for deploying smaller models with sophisticated inference strategies to enhance problem-solving accuracy while maintaining computational efficiency.
- Demonstration that smaller models (e.g., Llemma-7B) can achieve competitive accuracy to larger models (e.g., Llemma-34B) with lower compute budgets.
- The paper opens up new research directions in compute-optimal inference for LLMs, which is crucial for practical applications of AI in resource-constrained environments.

Weaknesses:
- The paper could elaborate more on the broader implications of these findings for the field of AI and mathematical reasoning.
- Limited discussion on how REBASE might perform in domains beyond mathematical reasoning tasks.

---

### Decision · Program_Chairs · 2024-10-09

Accept